# Exploring the Boundaries of GPT-4 in Radiology

Qianchu Liu[1], Stephanie L. Hyland[1], Shruthi Bannur[1], Kenza Bouzid[1],
Daniel C. Castro[1], Maria Teodora Wetscherek[1], Robert Tinn[1],
Harshita Sharma[1], Fernando Pérez-García[1], Anton Schwaighofer[1],
Pranav Rajpurkar[2], Sameer Tajdin Khanna[2], Hoifung Poon[1], Naoto Usuyama[1],
Anja Thieme[1], Aditya Nori[1], Matthew P. Lungren[1], Ozan Oktay[1] Javier Alvarez-Valle[1*]

[1] Microsoft Health Futures    [2] Harvard University

## Abstract

The recent success of general-domain large language models (LLMs) has significantly changed the natural language processing paradigm towards a unified foundation model across domains and applications. In this paper, we focus on assessing the performance of GPT-4, the most capable LLM so far, on the text-based applications for radiology reports, comparing against state-of-the-art (SOTA) radiology-specific models. Exploring various prompting strategies, we evaluated GPT-4 on a diverse range of common radiology tasks and we found GPT-4 either outperforms or is on par with current SOTA radiology models. With zero-shot prompting, GPT-4 already obtains substantial gains ($\approx 10\%$ absolute improvement) over radiology models in temporal sentence similarity classification (accuracy) and natural language inference ($F_1$). For tasks that require learning dataset-specific style or schema (e.g. findings summarisation), GPT-4 improves with example-based prompting and matches supervised SOTA. Our extensive error analysis with a board-certified radiologist shows GPT-4 has a sufficient level of radiology knowledge with only occasional errors in complex context that require nuanced domain knowledge. For findings summarisation, GPT-4 outputs are found to be overall comparable with existing manually-written impressions.

## 1 Introduction

Recently, the emergence of large language models (LLMs) has pushed forward AI performance in many domains; with many GPT-4 (OpenAI, 2023) powered applications achieving and even surpassing human performance in many tasks (Bubeck et al., 2023; Nori et al., 2023). There is a shift in paradigm towards using a unified general-domain foundation LLM to replace domain- and task-specific models. General-domain LLMs enable

a wider range of customised tasks without the need to extensively collect human labels or to perform specialised domain training. Also, with off-the-shelf prompting, applying LLMs is easier than the traditional training pipeline for supervised models.

While contemporary studies (Nori et al., 2023; Ranjit et al., 2023; Bhayana et al., 2023a) have started to explore the use of GPT-4 in the clinical domain, the readiness of GPT-4 in the radiology workflow remains to be rigorously and systematically tested. In this study, we set out the following research questions: (1) How can we evaluate GPT-4 on its ability to process and understand radiology reports? (2) How can we apply common prompting strategies for GPT-4 across different radiology tasks? (3) How does GPT-4 compare against SOTA radiology-specific models?

To answer these questions, we established a rigorous evaluation framework to evaluate GPT-4 on a diverse range of common radiology tasks including both language understanding and generation. The evaluation covers sentence-level semantics (natural language inference, sentence similarity classification), structured information extraction (including entity extraction, disease classification and disease progression classification), and a direct application of findings summarisation. We explored various prompting strategies including zero-shot, few-shot, chain-of-thought (CoT)(Wei et al., 2022), example selection (Liu et al., 2022), and iterative refinement (Ma et al., 2023), and we further experimented with adding self-consistency (Wang et al., 2023) and asking GPT-4 to defer handling uncertain cases to improve the reliability of GPT-4. For each task, we benchmarked GPT-4 with prior GPT-3.5 models (`text-davinci-003` and ChatGPT) and the respective state-of-the-art (SOTA) radiology models. Apart from reporting metric scores, we performed extensive qualitative analysis with a board-certified radiologist to understand the model errors by categorising them as ambiguous, label noise, or gen-

---
*Corresponding author: jaalvare@microsoft.com

uine model mistakes. We highlight the particular importance of qualitative analysis for open-ended generation tasks such as findings summariastion where GPT-4 may provide alternative solutions.

To sum up, our key contributions and *findings (in italics)* are:

1. **Evaluation Framework:** We proposed an evaluation and error analysis framework to benchmark GPT-4 in radiology. Collaborating with a board-certified radiologist, we pinpointed the limitations of GPT-4 and the current task paradigms, directing future evaluation pursuits to tackle more intricate and challenging real-world cases and to move beyond mere metric scores.

   *GPT-4 shows a significant level of radiology knowledge. The majority of detected errors are either ambiguous or label noise, with a few model mistakes requiring nuanced domain knowledge. For findings summarisation, GPT-4 outputs are often comparable to existing manually-written impressions.*

2. **Prompting Strategies:** We explored and established good practices for prompting GPT-4 across different radiology tasks.

   *GPT-4 requires minimal prompting (zero-shot) for tasks with clear instructions (e.g. sentence similarity). However, for tasks needing comprehension of dataset-specific schema or style (e.g. findings summarisation), which are challenging to articulate in instructions, GPT-4 demands advanced example-based prompting.*

3. **GPT-4 vs. SOTA:** We compared GPT-4 performance with task-specific SOTA radiology models for understanding and validating the paradigm shift towards a unified foundation model in the specialised domains.

   *GPT-4 outperforms or matches performance of task-specific radiology SOTA.*

## 2 Related Work

There have been extensive efforts to benchmark and analyse LLMs in the general-domain. Liang et al. (2023) benchmarks LLMs across broad NLP scenarios with diverse metrics. Hendrycks et al. (2021) measures LLMs' multitask accuracy across disciplines. Zheng et al. (2023) explores using LLMs as judge for open-ended questions. Bubeck et al.

(2023) further tests GPT-4's capabilities beyond language processing towards general intelligence (AGI), exploring tasks such as mathematical problem solving and game playing. Many other studies focus on testing specific capabilities such as reasoning from LLMs (Liu et al., 2023b; Espejel et al., 2023).

The evaluation of GPT-4 has also begun to garner interest in the medical field. For example, Lee et al. (2023) discusses the potential advantages and drawbacks of using GPT-4 as an AI chatbot in the medical field. Cheng et al. (2023) investigates possible applications of GPT-4 in biomedical engineering. Nori et al. (2023) evaluates GPT-4 for medical competency examinations and shows GPT-4 performance is well above the passing score. There have also been a few recent studies that evaluate GPT-4 in the radiology domain: Bhayana et al. (2023a,b) show that GPT-4 significantly outperforms GPT-3.5 and exceeds the passing scores on radiology board exams. Other studies have shown great potential from GPT-4 in various radiology applications such as simplifying clinical reports for clinical education (Lyu et al., 2023), extracting structures from radiology reports (Adams et al., 2023), natural language inference (NLI) (Wu et al., 2023b), and generating reports (Ranjit et al., 2023). While most of these studies focus on a specific application, our study aims for an extensive evaluation to compare GPT-4 against SOTA radiology models, covering diverse tasks and various prompting techniques.

Beyond prompting GPT-4, continued efforts are being made to adapt LLMs to the medical domain via fine-tuning. Med-PaLM and Med-PaLM-2 (Singhal et al., 2022, 2023) improve over PaLM (Chowdhery et al., 2022) and PaLM-2 (Anil et al., 2023) with medical-domain fine-tuning. Yunxiang et al. (2023) and Wu et al. (2023a) further fine-tune the open-source LLaMA model (Touvron et al., 2023) with medical-domain data. Van Veen et al. (2023) adapts LLMs to radiology data with parameter efficient fine-tuning. While these models offer lightweight alternatives, our study focuses on GPT-4 as it is still by far the best-performing model across many domains and represents the frontier of artificial intelligence (Bubeck et al., 2023).

## 3 Evaluation Framework

### 3.1 Task selection[1]

We benchmark GPT-4 on seven common text-only radiology tasks (Table 1) covering both understanding and generation tasks. The two sentence similarity classification tasks and NLI both require the understanding of sentence-level semantics in a radiology context, with NLI additionally requiring reasoning and logical inference. Structured information extraction tasks (disease classification, disease progression classification, and entity extraction) require both superficial entity extraction and inference from cues with radiology knowledge (e.g. 'enlarged heart' implies 'cardiomegaly'). For entity extraction, the model must further follow the schema-specific categorisation of entities. Finally, we evaluate GPT-4 on an important part of the radiology workflow: findings summarisation, i.e. condensing detailed descriptions of findings into a clinically actionable impression. These tasks cover different levels of text granularity (sentence-level, word-level, and paragraph-level) and different aspects of report processing, and hence give us a holistic view of how GPT-4 performs in processing radiology reports.

### 3.2 Prompting strategies

Alongside GPT-4 (`gpt-4-32k`), we evaluated two earlier GPT-3.5 models: `text-davinci-003` and ChatGPT (`gpt-35-turbo`). Model and API details are in Appendix A. For each task, we started with zero-shot prompting and progressively increased prompt complexity to include random few-shot (a fixed set of random examples), and then similarity-based example selection (Liu et al., 2022). For example selection, we use OpenAI's general-domain `text-embedding-ada-002` model to encode the training examples as the candidate pool to select $n$ nearest neighbours for each test instance. For NLI, we also explored CoT, as it was shown to benefit reasoning tasks (Wei et al., 2022). For findings summarisation, we replicated ImpressionGPT (Ma et al., 2023), which adopts dynamic example selection and iterative refinement.

To test the stability of GPT-4 output, we applied self-consistency (Wang et al., 2023) for sentence similarity, NLI, and disease classification. We report mean and standard deviation across five runs of

GPT-4 with temperature zero[2] and self-consistency results with majority voting (indicated by 'SC'). All prompts are presented in Appendix C.

### 3.3 Error analysis with radiologist

The authors did a first pass of the error cases to review easy instances requiring only general syntactic and linguistic knowledge (e.g. 'increased pleural effusion' versus 'decreased pleural effusion'). We then surfaced the cases where radiology expertise is required to a board-certified radiologist for a second-round review and feedback. For interpretability, we prompted GPT-4 to give an explanation after its answer. Reviewing both model answer and reasoning, we categorise each error into: ambiguous[3], label noise[4], or genuine mistake.

## 4 Experiments

### 4.1 Sentence similarity classification

**Task and model setup** In this task, the model receives as input a sentence pair and must classify the sentences as having the same, or different meanings. We evaluate the models on two sub-tasks: temporal sentence similarity classification (MS-CXR-T (Bannur et al., 2023b)) and RadNLI-derived sentence similarity classification. Temporal sentence similarity focuses on temporal changes of diseases. For RadNLI, we follow Bannur et al. (2023a) to use the subset of bidirectional 'entailment' and 'contradiction' pairs and discard the 'neutral' pairs to convert RadNLI (Miura et al., 2021) to a binary classification task.

The radiology SOTA for this task is BioViL-T (Bannur et al., 2023a) (a radiology-specific vision-language model trained with temporal multi-modal contrastive learning). The GPT performance is obtained from zero-shot prompting.

**Results** As shown in Table 2, all the GPT models outperform BioViL-T, achieving new SOTA. In particular, GPT-4 significantly outperforms both `text-davinci-003` and ChatGPT on MS-CXR-T, indicating an advanced understanding of disease progression. Error analysis revealed the majority of the GPT-4 (SC) errors are either ambiguous or

---

[1]The majority of our test set comes from data with restricted access (e.g., MIMIC (Johnson et al., 2019)).

[2]The OpenAI API for GPT-4 is non-deterministic even with temperature 0. We also explored varying the temperature parameter and found no improvement.

[3]An ambiguous case is when both GPT-4 error output and gold label can arguably be correct under different interpretations of the labels. For an example, an uncertain pathology could be labelled as 'presence' or 'absence'.

[4]the label is wrong and model is correct

Table 1: Results overview. GPT-4 either outperforms or is on par with previous SOTA. New SOTA is established by GPT-4 on sentence similarity and NLI (absolute improvement for accuracy and $F_1$ are reported). GPT-4 achieves near-ceiling performance in many tasks with < 1% mistake rate (shaded). ImpressionGPT (Ma et al., 2023) requires example selection and iterative example refinement.

| Task | Test samples | Prompting GPT-4 | GPT-4 performance | Mistake rate |
|---|---|---|---|---|
| Temporal sentence similarity | 361 | Zero-shot | New SOTA (↑10% acc.) | 0.0% |
| Sentence similarity (RadNLI) | 145 | Zero-shot | New SOTA (↑3% acc.) | 0.7% |
| Natural language inference (RadNLI) | 480 | Zero-shot + CoT | New SOTA (↑10% $F_1$) | 5.8% |
| Disease progression | 1326 | Zero-shot | On par with SOTA | 0.4% |
| Disease classification | 1955 | 10-shot* | On par with SOTA | 0.3% |
| Entity extraction | 100 | 200-shot* | On par with SOTA | – |
| Findings summarisation | 1606 / 576[†] | ImpressionGPT | On par with SOTA | – |

$n$-shot*: similarity-based example selection with $n$ examples; Mistake rate[5] = [# genuine mistakes] / [# test samples];
†: [MIMIC] / [Open-i]

label noise with only 1 model mistake in RadNLI (see Appendix B.1), indicating GPT-4 is achieving near-ceiling performance in these tasks.

Table 2: Zero-shot GPT-4 and GPT-3.5 achieve new SOTA (accuracy) on sentence similarity tasks. To test the consistency of GPT-4, we report mean and std. across five runs, and the self-consistency results ('SC').

| Model | MS-CXR-T | RadNLI |
|---|---|---|
| text-davinci-003 | 90.3 | 91.0 |
| ChatGPT | 92.0 | **95.2** |
| GPT-4 | **97.3 ± 0.2** | 94.1 ± 0.4 |
| GPT-4 (SC) | 97.2 | 93.8 |
| BioViL-T (Bannur et al., 2023a) | 87.8 | 90.5 |

## 4.2 Natural language inference (NLI)

**Task and model setup** We assess GPT on the original RadNLI classification dataset (Miura et al., 2021). The model receives input 'premise' and 'hypothesis' sentences, and determines their relation: one of 'entailment', 'contradiction', or 'neutral'.

We present GPT performance with zero-shot prompting and CoT. We compare GPT models against the current SOTA, a radiology-adapted T5 model (DoT5) which was trained on radiology text and general-domain NLI data (Liu et al., 2023a).

**Results** Table 3 shows that GPT-4 with CoT achieves a new SOTA on RadNLI, outperforming DoT5 by 10% in macro $F_1$. Whereas NLI has traditionally been a challenging task for earlier GPT models, GPT-4 displays a striking improvement. We also observe that CoT greatly helps in this task especially for GPT-3.5.

We further investigate how GPT-4 performs in cases that require different levels of radiology expertise[6], and we show that GPT-4 reaches the best performance in both generic and radiology-specific logical inference. CoT seems to help GPT models particularly to understand the radiology-specific cases. This is because CoT pushes the model to elaborate more on the radiology knowledge relevant to the input sentences, therefore giving sufficient context for a correct reasoning assessment (see Table B.4). Finally, we highlight that, even for GPT-4, there is still a gap in performance: the cases that specifically require radiology knowledge are more challenging than the other cases.

Table 3: GPT performance (macro $F_1$) on RadNLI with domain analysis. GPT-4 + CoT achieves new SOTA. Mean, std., and self-consistency ('SC') results are reported for GPT-4 + CoT across five runs.

| | All | need domain expertise? | |
| | | Yes | No |
|---|---|---|---|
| text-davinci-003 | 55.9 | 42.8 | 60.7 |
| + CoT | 64.9 | 54.1 | 68.4 |
| ChatGPT | 45.4 | 31.5 | 52.3 |
| + CoT | 70.5 | 65.6 | 70.2 |
| GPT-4 | 87.8 | 74.0 | 93.1 |
| + CoT | **89.3 ± 0.4** | **78.9 ± 1.4** | **93.5 ± 0.4** |
| + CoT (SC) | 89.2 | 78.8 | 93.6 |
| DoT5 (Liu et al., 2023a) | 79.8 | 70.1 | 86.4 |

## 4.3 Disease classification

**Task and model setup** The evaluation dataset is extracted from Chest ImaGenome (Wu et al., 2021) gold attributes on the sentence level. To fairly compare with the SOTA CheXbert (Smit et al., 2020) model, we focus on pleural effusion, atelectasis, pneumonia, and pneumothorax, which are common

---

[6]Our categorisation is based on Liu et al. (2023a)'s domain expertise annotations.

pathology names between CheXbert findings and Chest ImaGenome attributes. The output labels are 'presence' and 'absence' (binary classification) for each pathology. Detailed description of the label mapping is in Appendix D.

Besides the CheXbert baseline, we also include the silver annotations from Chest ImaGenome, produced by an ontology-based NLP tool with filtering rules (the Chest ImaGenome gold datasets are in fact human-verified silver annotations). To prompt GPT models, we started with zero-shot prompting, and then added 10 in-context examples with both random selection and similarity-based example selection. The example candidates are from the Chest ImaGenome silver data.

**Results** As shown in Table 4, there is progressive improvement from `text-davinci-003` to Chat-GPT and then to GPT-4. All the GPT models' zero-shot results outperform CheXbert. We are able to improve GPT-4 zero-shot performance with 10-shot random in-context examples. We achieve a further slight improvement with similarity-based example selection, approaching the performance of silver annotations.

We manually analysed the errors from the GPT-4 (*10) experiment and found that most (20 out of 30) are ambiguous, with the pathology cast as potentially present, rather than being easily labelled as present or not. This is particularly the case for pneumonia whose presence is typically only suggested by findings in the chest X-ray (See examples of such uncertain cases in Table B.6). The rest of the model errors are 5 cases of label noise and 5 model mistakes. With <1% mistake rate, GPT-4 is approaching ceiling performance in this task.

**Defer from uncertain cases** Given the large amount of uncertain and ambiguous cases in the dataset, we experimented with asking the model to output 'uncertain' alongside the presence and absence labels, and defer from these uncertain cases.[7] Table 5 shows that GPT-4 achieves very strong performance on those cases for which it is not uncertain. Note that pneumonia classification is dramatically improved and many positive cases of pneumonia are deferred. This aligns with our observation from the dataset that pneumonia is often reported as a possibility rather than a certain presence. We further test the robustness of GPT-4 in

this setup and report mean, standard deviation and majority vote results in Table E.1.

Table 4: GPT performance on Chest ImaGenome disease classification.

| Model | Micro $F_1$ | Macro $F_1$ |
| --- | --- | --- |
| `text-davinci-003` | 79.2 | 79.9 |
| ChatGPT | 89.7 | 85.0 |
| GPT-4 | 93.0 | 91.5 |
| GPT-4 (10) | 96.6 | 96.6 |
| GPT-4 (*10) | **97.9** | 97.5 |
| CheXbert | 73.6 | 73.1 |
| Silver | 97.8 | **98.9** |

(n): number of random shots; *: similarity-based example selection; Silver: Chest ImaGenome silver annotations.

Table 5: Zero-shot GPT-4 performance after deferring from uncertain cases on Chest ImaGenome dataset: GPT-4 (defer). Its performance is significantly improved from zero-shot GPT-4 (with binary output).

| | GPT-4 (defer) | GPT-4 |
| --- | --- | --- |
| Macro $F_1$ | 97.4 | 93.0 |
| Micro $F_1$ | 98.6 | 91.5 |
| Pleural effusion | 98.5 [103] | 95.3 [176] |
| Atelectasis | 99.0 [154] | 97.8 [233] |
| Pneumonia | 92.3 [16] | 75.7 [111] |
| Pneumothorax | 100.0 [17] | 97.3 [18] |

[n]: number of positive instances for each pathology.

## 4.4 RadGraph entity extraction

**Task and model setup** This task requires a model to extract observation and anatomy entities from radiology reports and determine their presence (present, absent, or uncertain) following the RadGraph schema (Jain et al., 2021). To evaluate the extraction, we report micro $F_1$ score counting a true positive when both the extracted entity text and the label are correct. RadGraph provides two datasets: MIMIC (Johnson et al., 2019) with both train and test data, and CheXpert (Irvin et al., 2019) (with only test data).

We compare with the SOTA RadGraph Benchmark model reported in Jain et al. (2021), which is based on DyGIE++ (Wadden et al., 2019) with Pub-MedBERT initializations (Gu et al., 2021). Regarding prompting strategy, we started with a randomly selected 1-shot example,[8] and then increased the number of random shots to 10. To push the performance, we leveraged the maximum context window

---

[7]This can be seen as an alternative way to allow for uncertainties compared with using the class logits (Nori et al., 2023) as the class logits are not available via the API endpoints.

[8]We first experimented with zero-shot prompting, which resulted in many output formatting errors. Adding one example resolved the formatting issue.

Table 6: GPT performance (micro $F_1$) on RadGraph entity extraction.

| Model | MIMIC | CheXpert |
|---|---|---|
| text-davinci-003 (1) | 56.2 | 49.2 |
| text-davinci-003 (10) | 83.2 | 79.5 |
| ChatGPT (1) | 47.1 | 42.2 |
| ChatGPT (10) | 70.6 | 67.5 |
| GPT-4 (1) | 36.6 | 25.3 |
| GPT-4 (10) | 88.3 | 84.7 |
| GPT-4 (200) | 91.5 | 88.4 |
| GPT-4 (*200) | 92.8 | **90.0** |
| RadGraph Benchmark | **94.3** | 89.5 |

(n): number of random shots; *: similarity-based example selection

of GPT-4, incorporating 200-shot examples with both random selection and similarity-based selection. Additionally, we found it is helpful to perform GPT inference on individual sentences before combining them for report-level output. The in-context examples are also on the sentence level (200-shot sentences roughly corresponds to 40 reports) from the train set.

**Results** As shown in Table 6, examples are crucial for GPT to learn this task. We observe a massive jump in performance when increasing the number of examples in the context. GPT-4 with 200 selected examples achieves overall on-par performance with RadGraph benchmark: while GPT-4 (*200) underperforms the RadGraph model on the in-domain MIMIC test set, GPT-4 surpasses RadGraph Benchmark on the out-of-domain CheXpert dataset. This indicates GPT-4 could be a more robust choice to generalise to out-of-domain datasets. Our error analysis reveals the errors are mostly due to GPT-4 failing to learn the schema specifics (Appendix B.5). For example, GPT-4 may extract the whole compound word ('mild-to-moderate') as the observation term, while the gold annotations break the word down ('mild' and 'moderate').

## 4.5 Disease progression classification

**Task and model setup** We evaluate on the temporal classification task from MS-CXR-T (Bannur et al., 2023b), which provides progression labels for five pathologies (consolidation, edema, pleural effusion, pneumonia, and pneumothorax) across three progression classes ('improving', 'stable', and 'worsening'). In this experiment, the input is the radiology report and the outputs are disease progression labels. We report macro accuracy for each

pathology due to class imbalance. As MS-CXR-T labels were originally extracted from Chest ImaGenome, we can also use Chest ImaGenome silver annotations as our baseline. We report GPT performance with zero-shot prompting.

**Results** Table 7 shows that there is again a large jump of performance from GPT-4 compared with the earlier GPT-3.5 models. Zero-shot GPT-4 achieves >95% across all pathologies and is comparable with Chest ImaGenome silver annotation. Our error analysis reveals that the majority of model errors are either label noise or ambiguous and the small mistake rate (0.4%) reflects the task is nearly solved.

Table 7: GPT performance on MS-CXR-T disease progression (macro accuracy).

| Model | Pl. eff. | Cons. | PNA | PTX | Edema |
|---|---|---|---|---|---|
| text-davinci-003 | 92.1 | 91.8 | 90.0 | 96.1 | 93.6 |
| ChatGPT | 91.0 | 84.8 | 84.5 | 93.0 | 89.8 |
| GPT-4 | **98.7** | **95.7** | 96.4 | 99.4 | 96.8 |
| Silver | 98.1 | 91.8 | **96.6** | **100.0** | **97.6** |

PNA: pneumonia; PTX: pneumothorax; Pl. eff.: pleural effusion; Cons.: consolidation; Silver: Chest ImaGenome silver annotations.

## 4.6 Findings summarisation

**Task and model setup** The findings summarisation task requires the model to summarise the input findings into a concise and clinically actionable impression section. We evaluate on the MIMIC (Johnson et al., 2019) and Open-i (Demner-Fushman et al., 2016) datasets and follow Ma et al. (2023) to report results on the official MIMIC test set and a random split (2400:576 for train:test) for Open-i. For metrics, we report RougeL (Lin, 2004) and the CheXbert score (Smit et al., 2020) (a radiology-specific factuality metric). We further conduct a qualitative comparison study on GPT-4 outputs.

For prompting strategies, we started with zero-shot and increased the number of random in-context examples to 10-shot. For GPT-4, we tried adding 100 examples with random selection and similarity-based selection. Examples are drawn from the respective train set for each dataset. We also replicated ImpressionGPT (Ma et al., 2023) with ChatGPT and GPT-4. ImpressionGPT performs dynamic example selection based on CheXbert labels and iteratively selects good and bad examples as in-context examples (The implementation details are found in Appendix G).

We compare with the previous supervised SOTA for this task (Hu et al., 2022) (which adopts a graph encoder to model entity relations from findings), as well as with DoT5 (Liu et al., 2023a), a strong zero-shot summarisation baseline.

**Results** While zero-shot GPT models all outperform DoT5, we observe that providing examples is crucial for this task: there is consistent and substantial improvement when increasing the number of in-context examples for all GPT models. A further boost can be achieved when we enable example selection for GPT-4 (*100). The more advanced ImpressionGPT brings the best performance out of GPT-4 and achieves performance comparable with the supervised SOTA.

**Qualitative comparison** To understand the differences between GPT-4 output and the manually-written impressions, we chose a random sample of reports and asked a radiologist to compare existing manually-written impressions with GPT-4 (ImpressionGPT) output. Table 9 demonstrates that for the majority of the cases ($\approx 70\%$), GPT-4 output is either preferred or comparable with the manually-written impression. Tables B.8 and B.9 show examples where GPT-4 outputs are more faithful to the findings than the manually-written impressions.

Table 8: GPT performance on findings summarisation. ImpressionGPT iteratively refines good and bad examples as in-context examples.

|  | MIMIC | | Open-i | |
| --- | --- | --- | --- | --- |
| **Model** | **R.** | **CB.** | **R.** | **CB.** |
| `text-davinci-003` | 22.9 | 41.8 | 14.5 | 41.9 |
| `text-davinci-003` (10) | 29.1 | 43.0 | 40.5 | 42.0 |
| ChatGPT | 20.0 | 40.5 | 14.8 | 39.6 |
| ChatGPT (10) | 31.0 | 42.5 | 40.6 | 41.0 |
| GPT-4 | 22.5 | 39.2 | 18.0 | 39.3 |
| GPT-4 (10) | 28.5 | 44.2 | 42.5 | 44.9 |
| GPT-4 (100) | 30.9 | 44.7 | 44.2 | 45.0 |
| GPT-4 (*100) | 38.4 | 47.4 | 59.8 | **47.3** |
| ChatGPT (ImpressionGPT) | 44.7 | 63.9 | 58.8 | 44.8 |
| GPT-4 (ImpressionGPT) | **46.0** | **64.9** | **64.6** | 46.5 |
| Hu et al. (2022) | **47.1** | 54.5 | 64.5 | – |
| DoT5 (Liu et al., 2023a) | – | – | 11.7 | 25.8 |

(n): number of random shots; *: similarity-based example selection; R.: RougeL; CB.: CheXbert.

Table 9: Percentage (%) with which the GPT-4 (ImpressionGPT) generated impression is equivalent or preferred compared with an existing manually-written one according to a radiologist.

| Sample ($n$) | **Manual Imp. preferred** | **Equiv.** | **GPT-4 preferred** | **Ambig.** |
| --- | --- | --- | --- | --- |
| Open-i (80) | 28.8 | 43.8 | 26.3 | 1.3 |
| MIMIC (40) | 25.0 | 10.0 | **57.5** | 7.5 |

Equiv.: equivalent; Ambig.: ambiguous;
Manual Imp.: Existing manual impression

## 5 Discussion

### 5.1 Error analysis and GPT-4 consistency

Moving beyond quantitative scores, we manually reviewed all GPT-4 errors in all the tasks (A detailed analysis is shown in Appendix B). We further analysed the consistency of the errors for a selection of tasks and reported the error breakdown in Table 10. We found the majority of the errors are either ambiguous or label noise. As an example of ambiguity, GPT-4 is extremely strict in identifying paraphrases and argues that one sentence contains minor additional information or slightly different emphasis. In fact, for sentence similarity, disease progression, and disease classification tasks, the model mistakes are < 1% of the test set (Table 1). We believe GPT-4 is achieving near-ceiling performance on these tasks. For entity extraction and findings summarisation, we found that GPT-4 output for many of the error cases is not necessarily wrong, but is offering an alternative to the schema or style in the dataset. This is verified by our qualitative analysis from Appendix B.5 and Section 4.6).

It is important to note that GPT-4 in our current study still makes occasional mistakes. Some mistakes are unstable across runs and can be corrected by self-consistency. Table 10 shows that GPT-4 is mostly consistent, and, for the few cases of inconsistent output, self-consistency can correct most of the model mistakes that occur in minority runs.[9] Another helpful strategy is to ask GPT-4 to defer when it is uncertain, as demonstrated by the disease classification experiments (Appendix B.3).

The remaining model mistakes are mostly cases where nuanced domain knowledge is required. For example, GPT-4 mistakenly equates 'lungs are hyperinflated but clear' with 'lungs are well-expanded

---

[9]Note that the overall scores from self-consistency experiments (Tables 2, 3 and E.1) do not reflect this quantitatively due to the noise from the many ambiguous cases.

Table 10: Self-consistency error analysis for GPT-4. Errors are categorised by whether they are consistent, occurring in minority runs (SC correct) or occurring in majority runs (SC incorrect). We further categorise errors into model mistakes and others (ambiguous or label noise). We observe the majority of the errors are consistent and many errors are not model mistakes. Within the cases of inconsistent output, self-consistency can correct most of the model mistakes. GPT-4 zero-shot performance is reported in this table (disease classification results are after we defer from the uncertain cases). Error breakdown for other single run experiments are in Table F.1.

| Task | Consistent | | SC correct | | SC incorrect | | |
| --- | --- | --- | --- | --- | --- | --- | --- |
| | Mistake | Other | Corrected mistake | Other | Mistake | Other | Total |
| Temporal sentence similarity | 0% | 72% | 10% | 0% | 0% | 18% | 11 |
| Sentence similarity (RadNLI) | 11% | 78% | 0% | 0% | 0% | 11% | 9 |
| RadNLI | 55% | 31% | 6% | 0% | 2% | 6% | 49 |
| Disease classification | 22% | 67% | 11% | 0% | 0% | 0% | 9 |
| All | 38% | 46% | 6% | 0% | 1% | 8% | 78 |

and clear' in MS-CXR-T. The former indicates an abnormality while the latter is describing normal lungs. We should point out that this mistake does not mean GPT-4 is fundamentally lacking the knowledge. In fact, when asked explicitly about it in isolation (e.g., difference between 'hyperinflated' and 'well-expanded lungs'), or when we reduce the complexity of the two sentences to 'lungs are hyperinflated' and 'lungs are well-expanded', GPT-4 is able to differentiate the two terms (Table B.3). We interpret it as nuanced radiology knowledge not being guaranteed to always surface for all contexts with all various prompts. While future prompting strategies might help with these cases, we must acknowledge that potential model mistakes cannot be fully ruled out. Therefore, a human in the loop is still required for safety-critical applications.

## 5.2 GPT-4 vs SOTA radiology models

Throughout the experiments, we first observed a significant jump of performance of GPT-4 compared with the prior GPT-3.5 (text-davinci-003 and ChatGPT), confirming the findings from previous studies (Nori et al., 2023). We then summarised the overall GPT-4 performance compared with radiology SOTA in Table 1. The key finding is that GPT-4 outperforms or is on par with SOTA radiology models in the broad range of tasks considered. We further notice that different tasks require different prompting efforts and strategies. For tasks such as sentence similarity, RadNLI, and disease progression, the task requirements can be clearly defined in the instruction. (For example, there is clear logical definition for 'entailment', 'neutral', and 'contradiction' in NLI). For such 'learn-by-instruction' tasks, a simple zero-shot prompting strategy for GPT-4 can yield significant gains

over task-specific baselines or nearly ceiling performance. Disease classification does not fall into this category due to the ambiguity in how to assign labels for the uncertain cases. Here, GPT-4 requires 10 examples to achieve comparable near-ceiling performance with previous SOTA. We show that zero-shot GPT-4 can also achieve near-ceiling performance if we defer from uncertain cases (Table 5) in this task. Another key point to note is that GPT-4 is a better choice than the previous SOTA Chest ImaGenome silver annotations for disease and disease progression classification, as the silver annotations are from rule-based systems that are not available to be re-used for other datasets.

Different from the above-mentioned tasks, it is not straightforward to articulate requirements in the instruction for entity extraction and findings summarisation. For entity extraction, the exact definition of observation and anatomy is schema-specific and in many cases can only be inferred from training examples. For findings summarisation, while there are general rule-of-thumb principles for writing a good impression, it is not possible to write down detailed instructions regarding the exact phrasing and style of the impressions in a particular dataset. We call these 'learn-by-example' tasks. Task-specific supervised models perform competitively on such tasks, as they can explicitly learn an in-domain distribution from all training examples. We found significant improvement of GPT models with increased number of examples compared with zero-shot, and GPT-4 with example selection can match supervised baselines. Future research can explore ways to combine GPT-4 and supervised models (e.g. treating the latter as plug-ins Shen et al. 2023; Xu et al. 2023).

## 6  Conclusion

This study evaluates GPT-4 on a diverse range of common radiology text-based tasks. We found GPT-4 either outperforms or is on par with task-specific radiology models. GPT-4 requires the least prompting effort for the 'learn-by-instruction' tasks where requirements can be clearly defined in the instruction. Our extensive error analysis shows that although it occasionally fails to surface domain knowledge, GPT-4 has substantial capability in the processing and analysis of radiology text, achieving near-ceiling performance in many tasks.

## 7  Limitations

In this paper, we focused on GPT-4 as it is the most capable and the best-performing LLM now across many domains and we would like to establish what best we can do with LLM in radiology. We leave it for future research to test and compare GPT-4 performance with other LLMs. In addition, as GPT-4 with the current prompting strategies in the study already achieves near-ceiling performance in many tasks, we leave an exhaustive experimentation of all existing prompting strategies for future research. For example, we have not explored the more recently proposed advanced prompting techniques including tree of thought (Yao et al., 2023) and self-critique (Shinn et al., 2023) and we encourage future research to apply techniques to help improve the reliability of GPT-4. Also, due to resource constraint, we did not perform self-consistency exhaustively for all tasks and for all GPT models. That being said, we believe the findings from this paper should already represent what an average user can get out of using GPT models on these tasks. The insights and learnings will be useful for designing future prompting strategies for radiology tasks, where particular tasks or error cases will require more prompting efforts.

Our error analysis shows that many of the existing radiology tasks contain intrinsic ambiguities and label noise and we call for more quality control when creating evaluation benchmarks in the future. Finally, our qualitative evaluation of the findings summarisation task is limited to a single radiologist. This is a subjective assessment that will be influenced by radiologist's own style and preference. The ideal scenario would be to ask radiologists who participated in the creation of the MIMIC or Open-i dataset to perform the assessment so that they have the same styling preference as the dataset. We are also planning to conduct more nuanced qualitative evaluation addressing different aspects of the summary in the future.

## 8  Ethical Considerations

we would like to assure the readers that the experiments in this study were conducted using Azure Open AI services which have all the compliance requirements as any other Azure Services. Azure Open AI is HIPAA compliant and preserves data privacy and compliance of the medical data (e.g., The data are not available to OpenAI). More details can be found in https:

//azure.microsoft.com/en-gb/resources/
microsoft-azure-compliance-offerings,
https://learn.microsoft.com/en-us/legal/
cognitive-services/openai/data-privacy
and https://learn.microsoft.com/
en-us/answers/questions/1245418/
hipaa-compliance. All the public datasets
used in this paper were also reviewed by MSR (Microsoft Research) IRB (OHRP parent organization
number IORG #0008066, IRB #IRB00009672)
under reference numbers RCT4053 and ERP10284.
IRB Decision: approved – Not Human Subjects Research (per 45§46.102(e)(1)(ii), 45§46.102(e)(5))

## Acknowledgments

We would like to thank the anonymous reviewers
and area chairs for their helpful suggestions. We
would also like to thank Hannah Richardson, Harsha Nori, Maximilian Ilse and Melissa Bristow for
their valuable feedback.

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

## A GPT model and API details

We evaluated GPT-4 (`gpt-4-32k`, ver. 0314), and two earlier GPT-3.5 models: `text-davinci-003` and ChatGPT (`gpt-35-turbo`, ver. 0301). We use Azure Cognitive Services API with the version '2023-03-15-preview'. Model names correspond to https://platform.openai.com/docs/models/overview. Regarding to GPT-4 pricing, at the time of conducting this study, it costs 0.06 USD per 1,000 tokens for prompt and 0.12 USD per 1,000 tokens. The actual cost depends on the task. For RadNLI with 361 samples, the cost is around 5 USD. The total cost of the evaluation conducted in this paper is around 5000 USD. We acknowledge that the cost of GPT-4 is high at the moment. As such, the findings from our paper can save the costs from future researchers who want to investigate similar research questions.

## B GPT-4 detailed error analysis

### B.1 Sentence similarity

We manually reviewed all GPT-4 errors and found that the errors are mostly ambiguous or label noise and these two tasks can be seen as nearly solved by GPT-4. For MS-CXR-T, the majority of the errors required identifying that 'improvement' was synonymous with 'decrease' for cases such as edema and lung opacity (all were ground truth paraphrase pairs) (See Table B.1). Among these cases, GPT-4 does recognise 'improvement' and 'decrease' describe positive change, but argues that 'improvement' and 'decrease' describe different aspects of the change. Confirmed with our radiologist, GPT-4 reasoning here is pedantic but is understandable.

In one error however, GPT-4 reasons that 'improved' and 'decrease' indicate opposite direction of change, which is a clear error showing the radiology-specific knowledge does not surface from GPT-4 in this case. Fortunately, this error case is only the minority across all runs (2 out of 5 cases) and is corrected by SC.

The finding is similar for the RadNLI sentence similarity task where the majority of the errors are ambiguous cases where GPT-4 flagged that one sentence contained slightly more information than the other, hence were not strict paraphrases. There is one genuine error where GPT-4 mistakenly equates 'lungs are hyperinflated but clear' with 'lungs are well-expanded and clear' (See Table B.2). The former indicates an abnormality while the latter is normal. We should point out that this behaviour does not mean GPT-4 is fundamentally lacking the knowledge. In fact, if we reduce the complexity of the two sentences to 'lungs are hyperinflated' and 'lungs are well-expanded', GPT-4 is able to differentiate between the two terms. Therefore, we interpret the error as the domain knowledge has not being surfaced in a complex context.

Self-consistency does not improve the overall results and most errors are consistent across runs. For MS-CXR-T, 8 out of the 11 error cases are consistent. For RadNLI sentence similarity, 8 out of 9 error cases are consistent across runs. For the inconsistent cases, apart from one genuine error from MS-CXR-T which is corrected by self-consistency, all inconsistent cases are ambiguous.

| Sentence 1 | interval improvement in mild pulmonary edema |
|---|---|
| Sentence 2 | interval decrease in mild pulmonary edema |
| GPT-4 reasoning | The first sentence describes an improvement in the mild pulmonary edema, which means the condition has gotten better. The second sentence describes a decrease in the mild pulmonary edema, which means the amount or severity of the edema has reduced. Although both sentences indicate a positive change, they describe different aspects of the change in the pulmonary edema. |
| Error category | ambiguous |
| Sentence 1 | opacities in the left base have minimally improved. |
| Sentence 2 | opacities in the left base have minimally decreased. |
| GPT-4 reasoning | The first sentence states that the opacities have improved, which means they have become less severe or less noticeable. The second sentence states that the opacities have decreased, which means they have reduced in size or number. Although both sentences describe a positive change, they convey different aspects of the change in opacities. |
| Error category | ambiguous |
| Sentence 1 | left basal consolidation has slightly improved. |
| Sentence 2 | left basal consolidation has slightly decreased. |
| GPT-4 reasoning | The first sentence states that the left basal consolidation has slightly improved, meaning the condition has gotten better. The second sentence states that the left basal consolidation has slightly decreased, which implies that the size or extent of the consolidation has reduced. Although both sentences indicate a positive change, they describe different aspects of the change. |
| Error category | ambiguous |
| Sentence 1 | mild improvement in right upper lung opacity, possibly a hematoma, status post right upper lobe resection. |
| Sentence 2 | mild decrease in right upper lung opacity, possibly a hematoma, status post right upper lobe resection. |
| GPT-4 reasoning | The two sentences describe different changes in the right upper lung opacity. Sentence 1 mentions a "mild improvement," while Sentence 2 mentions a "mild decrease." These changes are opposite in nature, with improvement suggesting a positive change and decrease suggesting a reduction in opacity. |
| Error category | genuine mistake |

Table B.1: Examples of GPT-4 errors on the MS-CXR-T temporal sentence similarity dataset. Note that all pairs are labelled (ground truth) as paraphrases, but were classified as different by GPT-4. In the first three examples, GPT-4 gives sound reasoning. The last example shows erroneous reasoning where the model thinks improvement and decrease indicate opposite directions of change, but fortunately the last example comes from 2 out of 5 runs and can be corrected by self-consistency.

| Sentence 1 | Osseous structures are without an acute abnormality. |
|---|---|
| Sentence 2 | The osseous structures are grossly unremarkable, although evaluation is limited secondary to body habitus. |
| GPT-4 output and reasoning | different. While both sentences describe the osseous structures as not having any significant issues, Sentence1 states that there are no acute abnormalities, while Sentence2 mentions that the evaluation is limited due to the patient's body habitus, implying that there might be some limitations in the assessment. |
| True label | paraphrase |
| Error category | ambiguous |
| Sentence 1 | Lungs are hyperinflated but clear. |
| Sentence 2 | Lungs are well-expanded and clear without focal consolidation concerning for pneumonia. |
| GPT-4 output and reasoning | paraphrase. Both sentences describe lungs that are clear and without any signs of pneumonia. The first sentence uses the term "hyperinflated" while the second uses "well-expanded," but both essentially convey the same meaning of lungs being expanded and clear. |
| True label | different |
| Error category | genuine mistake |

Table B.2: Examples of GPT-4 errors on the RadNLI-derived sentence similarity dataset. In the first example, the label is ambiguous and GPT-4 highlights the additional information in the second sentence. In the second example, GPT-4 erroneously equates hyperinflated (a pathological state) with 'well-expanded' lungs.

| Sentence 1 | Lungs are hyperinflated but clear. |
|---|---|
| Sentence 2 | Lungs are well-expanded and clear without focal consolidation concerning for pneumonia. |
| GPT-4 output and reasoning | paraphrase. Both sentences describe lungs that are clear and without any signs of pneumonia. The first sentence uses the term "hyperinflated" while the second uses "well-expanded," but both essentially convey the same meaning of lungs being expanded and clear. |
| True label | different |
| Sentence 1 | Lungs are hyperinflated. |
| Sentence 2 | Lungs are well-expanded. |
| GPT-4 output and reasoning | different. Although both sentences describe the state of the lungs, they convey different meanings. "Hyperinflated" implies that the lungs are excessively inflated, which could indicate a medical issue. On the other hand, "well-expanded" suggests that the lungs are in a healthy state, functioning properly with adequate inflation. |
| True label | different |

Table B.3: An example to show that domain knowledge is surfaced from GPT-4 when we reduce the complexity of the context as in the second example.

## B.2 RadNLI

We observe a higher error rate in NLI compared to sentence similarity. This can be explained by the subtle nature of the task that requires precise logical reasoning in order to correctly recognise textual entailment from neutral examples and contradictions. Out of the 45 model "errors" from GPT-4 + CoT (SC), we identified 13 logical errors where the model's reasoning indicated a lack of understanding of entailment as it appeared to look for paraphrases 'the hypothesis does not provide all information in the premise' or it confused the directionality of entailment 'the premise is part of the hypothesis'. Additionally, we found 18 label-noise examples for which GPT-4 was arguably right. An extra set of 14 errors were domain specific such as 'unchanged' does not imply 'normal' and 'enlarged' suggests 'not top normal' in a radiology context (See Table B.5). While most of the errors are consistent across run, there are 7 cases of inconsistent output across runs. For 4 cases, majority run does not align with the gold label but these cases are mostly label noise. For 3 cases, majority corrects inconsistent output and all 3 cases are genuine errors from minority runs.

We also show in Table B.4 the examples where CoT improves GPT-4 performance on RadNLI.

## B.3 Disease Classification

We analyse the errors for the GPT-4 (defer) setup after we defer from uncertain cases. We collect a total of 8 error cases for all pathologies from a single run: there are two cases of label noise, while the other errors are largely ambiguous and uncertain cases where GPT-4 should output 'uncertain' but outputs 'absence' instead. For example, in 'alternatively, could be due to infection, a toxic or allergic drug reaction or hemorrhage.', GPT-4 should output uncertain for pneumonia rather than absence. There are two genuine errors where GPT-4 fails to identify 'pleural effusion' from 'fluid did not accumulate acutely', and mistakenly assigns the absence label for pleural effusion in the sentence 'pleural effusion is nearly resolved'. When we compare the outupt before and after we add the 'uncertain' option to the model, we observe an overall improvement in $F_1$, many of the ambiguous errors from binary setup are now assigned with 'uncertain' label. In addition, some of the genuine errors from binary setup (e.g., not understanding 'linear opacities' indicates atelectasis) are assigned

with 'uncertain' as well as GPT-4 reasons that 'linear opacities' can possibly reflect atelectasis. This indicates that the model is able to defer in the case of an obvious erroneous output.

We test self-consistency for the GPT-4 (defer) set up across five runs, the 8 identified errors are consistent. There is an extra error from 1 out of 5 runs where the model makes a genuine mistake: GPT-4 thinks pleural effusion is present based on "there is no pneumothorax, large effusion, or congestion." Fortunately, this obvious error can be corrected by majority voting from self-consistency.

## B.4 Disease Progression

The error analysis reveals that GPT-4 is achieving ceiling performance on this task. We manually reviewed a total of 34 errors from all the pathologies and found most of the model mismatches are either due to label noise (7) or the case being ambiguous (21). Many of the ambiguous cases appeared where the pathologies mentioned are uncertain in the first place and therefore harder to assess for change. The genuine errors are either because the model fails to recognise pathologies (e.g. 'hemorrhage' will indicate consolidation) or failed to associate the change word with the pathology due to unfamiliarity with radiology-specific phrasing (e.g., not able to recognise edema is worsening from 'increased opacities consistent with mild pulmonary edema').

Table B.7 shows the example errors from GPT-4 on disease progression.

## B.5 RadGraph

We examine the GPT-4 (*200) output on MIMIC and collect in total 137 error cases. We categorise these errors into 31 cases where gold entities are not identified from predictions, and 73 cases where predicted entities are not found in gold, and 33 cases where the entity tokens are extracted but the labels are not correct. For the missing gold entities, we found that most of the cases (22 out of 33) are due to GPT breaking up the word differently than the gold annotations: e.g., While GPT-4 extracts the whole phrase 'mild-to-moderate' as one observation entity, the gold annotation extracts 'mild' and 'moderate' separately. For the 73 over-predicted entities, we found that the majority of the mismatch is due to inconsistency in the annotation. For example, in the sentence 'an esophageal drainage tube passes into the stomach and out of view' GPT extracts 'out of view' as the observation but the gold annotation does not. However, in

| | |
|---|---|
| Premise | Mediastinal and hilar contours are grossly unremarkable. |
| Hypothesis | Bony structures are unremarkable. |
| GPT-4 + CoT | REASON: The premise talks about the mediastinal and hilar contours being unremarkable, which refers to the structures in the chest area. The hypothesis talks about bony structures being unremarkable, which refers to the bones. These two statements are not directly related to each other, as one talks about chest structures and the other talks about bones. Therefore, there is no clear evidence to support or contradict the hypothesis based on the premise. Hence, the relation between the premise and hypothesis is neutral.
ANSWER: neutral |
| GPT-4 | ANSWER: entailment |
| True label | neutral |
| Premise | The lungs are clear. |
| Hypothesis | The lungs are well expanded. |
| GPT-4 + CoT | Both the premise and hypothesis are related to the lungs, but they express different information. The premise states that the lungs are clear, which means that there are no signs of disease or infection. On the other hand, the hypothesis states that the lungs are well expanded, which means that they are functioning properly and have good air flow. These two statements are not contradictory, but they do not necessarily entail each other either. It is possible for the lungs to be clear but not well expanded, or for the lungs to be well expanded but not necessarily clear. Therefore, the relation between the premise and hypothesis is neutral.
ANSWER: neutral |
| GPT-4 | ANSWER: entailment |
| True label | neutral |

Table B.4: Examples where CoT improves GPT-4 performance on RadNLI.

| | |
|---|---|
| Premise | The heart is mildly enlarged, unchanged. |
| Hypothesis | The heart is top normal in size, which is exaggerated by the low lung volumes. |
| GPT-4 output and reasoning | REASON: The premise states that the heart is mildly enlarged and unchanged, while the hypothesis suggests that the heart is top normal in size and appears larger due to low lung volumes. These two statements provide different information about the heart size and do not support or contradict each other directly. ANSWER: neutral |
| True label | contradiction |
| Premise | The pulmonary vasculature is normal. |
| Hypothesis | There is no pulmonary edema. |
| GPT-4 output and reasoning | REASON: The premise states that the pulmonary vasculature is normal, which implies that there are no abnormalities in the blood vessels of the lungs. The hypothesis states that there is no pulmonary edema, which is a condition where fluid accumulates in the lungs due to issues with the blood vessels or heart. Since a normal pulmonary vasculature would not typically result in pulmonary edema, the hypothesis can be inferred from the premise. ANSWER: entailment |
| Error category | genuine mistake |
| True label | neutral |
| Error category | genuine mistake |
| Premise | Opacities involving the right middle lobe and left lower lobe are again demonstrated and may represent atelectasis or infection in the appropriate setting. |
| Hypothesis | Minimal left basilar opacity suggests minor atelectasis. |
| GPT-4 output and reasoning | REASON: The premise mentions opacities in the right middle lobe and left lower lobe, which could represent atelectasis or infection. The hypothesis, on the other hand, talks about minimal left basilar opacity suggesting minor atelectasis. While both sentences discuss opacities and atelectasis, they refer to different parts of the lungs. Therefore, the relation between the premise and hypothesis is undetermined. ANSWER: neutral |
| True label | entailment |
| Error category | genuine mistake |

Table B.5: GPT-4 errors on RadNLI. In the first example, GPT-3 misses that 'enlarged' entails 'not top-normal' for hear size. In the second example, GPT-4 fails to understand that normal pulmonary vasculature does not necessarily always indicates no normal pulmonary edema. In the last example, GPT-4 fails to understand left basilar opacity and opacities in the left lower lobe refer to the same location.

| Text | scattered vague opacities bilaterally are consistent with clinical history of septic emboli, unknown whether active or scar. |
|---|---|
| Gold | (pneumonia, absent) |
| GPT-4 answer and reasoning | (pneumonia, present). The text mentions "scattered vague opacities bilaterally" which are consistent with the clinical history of septic emboli. This suggests the presence of pneumonia. |
| Error category | ambiguous |
| Text | no large pleural effusions are seen, although trace effusions will be difficult to exclude. |
| Gold | (pleural effusion, absent) |
| GPT-4 answer and reasoning | (pleural effusion, present). The text mentions that no large pleural effusions are seen, but trace effusions are difficult to exclude, which implies that small pleural effusions might be present. |
| Error category | ambiguous |

Table B.6: Examples of ambiguous GPT-4 errors on the disease classification binary classification task from Chest ImaGenome Gold dataset.

| Text | ... Small focus of hemorrhage or atelectasis at the base of the postoperative right lung has cleared... |
|---|---|
| Gold | (consolidation, improving) |
| GPT-4 answer and reasoning | (consolidation, stable). Consolidation is not mentioned in the input text. The findings discussed include persistent moderate right pneumothorax, small bilateral pleural effusions, small focus of hemorrhage or atelectasis, and mild generalized pulmonary edema. |
| Error category | genuine mistake |
| Text | Increased interstitial opacity with mediastinal vascular fullness and Kerley B lines is consistent with mild to moderate pulmonary edema |
| Gold | (edema, worsening) |
| GPT-4 answer and reasoning | (edema, stable). The text does not provide any information about the change in the edema's condition compared to a previous state. Therefore, we cannot determine if it is improving or worsening, and we assume it is stable. |
| Error category | genuine mistake |

Table B.7: Examples of GPT-4 genuine mistakes on the disease progression classification task. In the first example, GPT-4 fails to identify that hemorrhage is a type of consolidation. In the second example, GPT-4 fails to understand the radiology-specific phrasing that the change 'consistent with edema will also apply to the edema.

another gold example in the training data: 'right internal jugular line ends in the right atrium and an upper enteric drain passes into a non-distended stomach and out of view .', 'out of view' was annotated as the observation. This reflects the intrinsic ambiguity of annotating what counts as observation and anatomies in the schema. In another 25 cases of over-predicted entities, GPT over-extracts words such as 'probable', 'definite' as observations. Finally, we found 33 cases where GPT-4 extracts the correct entities but assigns the wrong label, and they are mostly due to the confusion of uncertain labels. Many of these are ambiguous, for example, for the sentence 'No definite focal consolidation identified', the gold annotation assigns an 'observation-uncertain' label for 'consolidation' but GPT assigns 'observation-absent'.

This error analysis reveals the intrinsic challenge of learning schema-specific annotations in this task, as there may not be a single clear-cut standard in some cases. GPT output often is not wrong but is offering an alternative solution.

## B.6 Findings summarisation

Table B.9 and Table B.8 show examples where GPT-4 (ImpressionGPT) outputs are favoured than existing manually-written impressions.

| | |
|---|---|
| Findings | There is mild cardiomegaly. Mediastinal contours appear within normal limits. There are small bilateral pleural effusions, left greater than right with left basilar opacities. No pneumothorax. Mild degenerative changes of the thoracic spine. |
| Existing impression | Cardiomegaly with small bilateral pleural effusions and left basilar atelectasis. |
| GPT-4 impression | Mild cardiomegaly and small bilateral pleural effusions with left basilar opacities. |
| Findings | Cardiomegaly is noted. No pleural effusions. No pneumothorax. There is perihilar prominence and interstitial opacification. |
| Existing impression | Cardiomegaly and mild interstitial pulmonary edema. |
| GPT-4 impression | Stable cardiomegaly with prominent perihilar opacities which may represent scarring or edema. |

Table B.8: Examples where GPT-4 (ImpressionGPT) impressions are favoured over the existing manually-written impressions on Open-i dataset. In both examples, GPT-4 outputs have more faithful and more complete details to the findings.

| | |
|---|---|
| Findings | Single AP upright portable view of the chest was obtained. The patient's overlying chin obscures the medial bilateral upper lobes. The cardiac silhouette remains enlarged. Prominence of the pulmonary arteries is partially imaged and again seen. Evidence of diaphragmatic/pleural plaques is seen bilaterally suggesting prior asbestos exposure. |
| Existing impression | The patient's chin overlies the bilateral medial upper lobes, obscuring the view. Given this, the cardiac silhouette is persistently enlarged. There is again prominence of the pulmonary arteries. Pulmonary vascular congestion appears improved. |
| GPT-4 impression | Cardiomegaly and prominence of pulmonary arteries. Bilateral diaphragmatic/pleural plaques suggestive of prior asbestos exposure. |
| Findings | AP upright and lateral views of the chest provided. Vascular stent is seen in the region of the right brachiocephalic vein. The heart is moderately enlarged. There is mild interstitial pulmonary edema. Previously noted ET and NG tubes have been removed. No large pleural effusion. Mediastinal contour is stable. Bony structures are sclerotic which could reflect renal osteodystrophy. |
| Existing impression | Mild cardiomegaly with mild interstitial pulmonary edema. |
| GPT-4 impression | Moderate cardiomegaly with mild interstitial pulmonary edema. |

Table B.9: Examples where GPT-4 (ImpressionGPT) impressions are favoured over the existing manually-written impressions on MIMIC dataset The first example shows that the existing manually-written impressions written by radiologist are not always perfect and may contain irrelevant information (in orange). In the second example GPT-4 gives the correct detail ('Moderate cardiomegaly') based on the findings whereas the existing impression incorrectly concludes with 'mild cardiomegaly'.

# C  Prompts

For each task, We write the prompt to describe the task requirements with the input and output format, and we assign GPT the role of a radiologist in the system message. The prompts are written in one go without tweaking and tuning wording for each task.

## C.1  Sentence Similarity

The chat prompt for zero-shot sentence similarity classification is shown in Fig. C.1.

## C.2  RadNLI

Figure C.2 presents the zero-shot chat prompt for RadNLI.

## C.3  Disease Classification

Figure C.3 presents the zero-shot chat prompt for disease classification.

## C.4  RadGraph Entity Extraction

Figure C.4 shows the zero-shot chat prompt for RadGraph Entity Extraction.

## C.5  Disease Progression Prompt

Figure C.3 presents the zero-shot chat prompt for disease progression classification.

## C.6  Summarisation Prompt

Figure C.6 presents the zero-shot chat prompt for findings summarisation.

| | |
|---|---|
| System | You are a radiologist. Assess whether two sentences are describing the same meaning (paraphrase) or different meaning (different) regarding the change information. Reply with 'paraphrase' or 'different' first and then explain. |
| User | - - INPUT
Sentence1: Left lower lobe collapse stable.
Sentence2: Persistent left lower lobe collapse.<\|endofprompt\|>
ANSWER: |

Figure C.1: Zero-shot Chat Prompt for Sentence Similarity

| | |
|---|---|
| System | You are a radiologist performing natural language inference on 2 sentences: premise and hypothesis. You need to judge which following three relations hold for the premise and hypothesis:

entailment: The hypothesis can be inferred from the premise.
contradiction: The hypothesis can NOT be inferred from the premise.
neutral: The inference relation of the premise and the hypothesis is undetermined.

Given the input, compare premise and hypothesis and reply with the following structure:

REASON: <Text explaining the decision step by step>
ANSWER: <entailment \| neutral \| contradiction> |
| User | - - INPUT
Premise: There is no pleural effusion pneumothorax.
Hypothesis: No focal consolidation, pleural effusion, pneumothorax, or pulmonary edema.
What is the relation between premise and hypothesis? Explain your reason first and then answer entailment, neutral or contradiction.<\|endofprompt\|>
REASON: |

Figure C.2: Zero-shot Chat Prompt for RadNLI

| System | You are a radiologist that identifies findings from radiology reports on chest X-rays. Given a piece of radiology text in the following input format: |
|---|---|

System   You are a radiologist that identifies findings from radiology reports on
chest X-rays. Given a piece of radiology text in the following input format:

```
<INPUT>
<text>
</INPUT>
```

Assess the following findings from the input text: "Pleural Effusion",
"Atelectasis", "Pneumonia", "Pneumothorax". Answer "present" if the finding
is present; Answer "absent" if the finding is absent. Answer "not mentioned"
if the finding is not mentioned from text. Reply with a list of tuples and
then briefly explain following the format:

```
<OUTPUT>
ANSWER:
[
("Pleural Effusion", "present"|"absent"|"not mentioned"),
("Atelectasis", "present"|"absent"|"not mentioned"),
("Pneumonia", "present"|"absent"|"not mentioned"),
("Pneumothorax", "present"|"absent"|"not mentioned"),
]
EXPLANATION: <explanation>
</OUTPUT>
```

User     
```
<INPUT>
cardiomediastinal and hilar contours are unremarkable.
</INPUT>
```
Assess the requested findings from the above input text. Answer "present",
"absent" or "not mentioned" for each finding. Reply with a list of tuples
first and then briefly explain.<|endofprompt|>
```
<OUTPUT>
ANSWER:
```

Figure C.3: Zero-shot Chat Prompt for Disease Classification

| | |
|---|---|
| System | You are a radiologist performing clinical term extraction from the FINDINGS and IMPRESSION sections in the radiology report. Here a clinical term can be either anatomy or observation that is related to a finding or an impression. The anatomy term refers to an anatomical body part such as a 'lung'. The observation terms refer to observations made when referring to the associated radiology image. Observations are associated with visual features, identifiable pathophysiologic processes, or diagnostic disease classifications. For example, an observation could be 'effusion' or description phrases like 'increased'. You also need to assign a label to indicate whether the clinical term is present, absent or uncertain. Given a piece of radiology text input in the format:

`<INPUT>`
`<text>`
`</INPUT>`

reply with the following structure:

`<OUTPUT>`
ANSWER: tuples separated by newlines. Each tuple has the format: (`<clinical term text>`, `<label: observation-present |observation-absent|observation-uncertain|anatomy-present>`). If there are no extraction related to findings or impression, return ()
`</OUTPUT>` |
| Example user | `<INPUT>`
No convincing evidence of pneumothorax or pneumomediastinum .
`</INPUT>`

What are the clinical terms and their labels in this text? Discard sections other than FINDINGS and IMPRESSION: eg. INDICATION, HISTORY, TECHNIQUE, COMPARISON sections. If there is no extraction from findings and impression, return ().

`<OUTPUT>`
ANSWER: |
| Example assistant | ('pneumothorax', 'observation-absent')
('pneumomediastinum', 'observation-absent')
`</OUTPUT>` |

Figure C.4: 1-shot Chat Prompt for RadGraph Entity Extraction

| System | You are a radiologist that identifies the progression of pathologies from radiology text. Given a radiology report in the following input format: |
|---|---|

```
<INPUT>
<text>
</INPUT>
```

Assess the following findings from the input text: "Edema". Answer "improving" if the finding is improving. Answer "worsened" if the finding is worsened. Answer "stable" if the finding is stable.Reply with a prediction and then briefly explain in the following format:

```
<OUTPUT>
ANSWER:
[
("Edema", "worsened"|"improving"|"stable"),
]
EXPLANATION: <explanation>
</OUTPUT>
```

| User | <INPUT> |
|---|---|

```
FINAL REPORT
INDICATION: Chest pain and bradycardia. Evaluate for pneumonia.
COMPARISONS: Chest radiograph from ___.
TECHNIQUE: A single AP upright view of the chest was obtained.
FINDINGS: Since prior exam, there are new interstitial opacities and vascular
congestion, most consistent with moderate pulmonary edema. There is no focal
airspace opacity, pleural effusion, or pneumothorax. The mediastinal contours
are normal. The heart size is mildly enlarged.
IMPRESSION: New moderate pulmonary edema.
</INPUT>
```

Assess the above input text. Answer "improving" if the finding is improving. Answer "worsened" if the finding is worsened. Answer "stable" if the finding is stable. Reply with a prediction and then briefly explain:<|endofprompt|>

```
<OUTPUT>
ANSWER:
```

Figure C.5: Zero-shot Chat Prompt for Disease Progression Classification

| System | You are a radiologist that can write an impression section in a radiology report. Given the findings section of the report as the input: |
|---|---|
| | `<INPUT>`
`<findings>`
`</INPUT>` |
| | generate impression section: |
| | `<OUTPUT>`
`IMPRESSION:<impression>`
`</OUTPUT>` |
| User | `<INPUT>`
Left PICC tip is seen terminating in the region of the distal left brachiocephalic vein. Tracheostomy tube is in unchanged standard position. The heart is moderately enlarged. Marked calcification of the aortic knob is again present. Mild pulmonary vascular congestion is similar. Bibasilar streaky airspace opacities are minimally improved. Previously noted left pleural effusion appears to have resolved. No pneumothorax is identified. Percutaneous gastrostomy tube is seen in the left upper quadrant.
`</INPUT>`
Generate the impression section based on the input findings:`<\|endofprompt\|>`
`<OUTPUT>`
`IMPRESSION:` |

Figure C.6: Zero-shot Chat Prompt for Findings Summarisation

## D Label mapping details for disease classification

The attributes in Chest ImaGenome are labelled as 'yes' or 'no'. When there are no such labels, we assign the CheXbert 'missing' class. When calculating the scores, we collapse 'missing' and 'no' labels into the negative class, and the 'yes' label is treated as the positive class. CheXbert predicts four labels for each pathology: 'present', 'absent', 'not mentioned/missing', and 'uncertain'. To conform with the Chest ImaGenome dataset, we combine 'present' and 'uncertain' into the positive class, and 'absent' and 'missing' into the negative class.

## E Self-consistency Results for Disease Classification

Table E.1 shows the self-consistency results for disease classification after deferring from uncertain cases.

|                    | GPT-4 (SC) | GPT-4 (mean)    |
|--------------------|------------|-----------------|
| Macro $F_1$        | 97.44      | $97.46 \pm 0.12$ |
| Micro $F_1$        | 98.56      | $98.56 \pm 0.12$ |
| Pleural Effusion   | 98.47      | $98.38 \pm 0.21$ |
| Atelectasis        | 98.99      | $98.38 \pm 0.21$ |
| Pneumonia          | 92.30      | $92.42 \pm 0.25$ |
| Pneumothorax (18)  | 100.0      | 100.0           |

Table E.1: mean, standard deviation and the self-consistency results for Zero-shot GPT-4 after deferring from uncertain cases

## F Error breakdown for single-run experiments

Table F.1 shows the error breakdown for single-run experiments.

## G Implementation details for ImpressionGPT

We replicated the latest ImpressionGPT (Ma et al., 2023) framework using both Chat-GPT(as proposed in the original work) and GPT-4. We reproduced this work based on the publicly available code `https://github.com/MoMarky/ImpressionGPT`. We set the hyperparameter values to match the optimal settings reported in the ablation study from the paper (Ma et al., 2023) which are different from the default values hard-coded in the repository. We therefore select $N_s = 15$ most similar in-context examples in the dynamic prompt. Additionally, we iteratively inject as many bad examples $B_d = n$ and update the single good example $G_d = 1$ with highest Rouge-1 score using Rouge-1 threshold $T = 0.7$. Finally, the iterative process is run for $I = 17$ iterations. We evaluate on the same test split shared in `https://github.com/MoMarky/radiology-report-extraction` for Open-i dataset and the official test split for MIMIC-CXR. We note a performance drop for ChatGPT (ImpressionGPT) baseline (Rouge-L=44.7 vs 47.93 for MIMIC-CXR and Rouge-L=58.8 vs 65.47 for Open-i) compared to the results reported in (Ma et al., 2023). The default hyperparameters in the repository also did not produce the expected results.

| Task | Mistake | Other | Total | Mistake Rate |
|------|---------|-------|-------|--------------|
| Disease Classification | 17% | 83% | 30 | 5 / 1955 (0.3%) |
| Disease Progression Classification | 18% | 82% | 34 | 13 / 1326 (0.4%) |

Table F.1: GPT-4 Error breakdown for single-run classification experiments: disease classification (*10) and disease progression (zero-shot). Errors are categories into mistakes and other (ambiguous or label noise).