# OpenReview forum: "Exploring the Boundaries of GPT-4 in Radiology"
_EMNLP/2023/Conference — EMNLP 2023 Main_

### Official Review · Reviewer_cPeU · 2023-07-23

**Soundness:** 4

**Excitement:**

4: Strong: This paper deepens the understanding of some phenomenon or lowers the barriers to an existing research direction.

**Paper Topic And Main Contributions:**

This paper conducted extensive experiments and analyses of GPT-4 in the radiology domain. Experimental results showed that GPT-4 has a strong performance when compared with SOTA baselines. In addition, extensive analysis with radiologists showed that GPT-4 has a sufficient level of specific knowledge.

**Reasons To Accept:**

[1] The paper is well-written. The analysis part, prompt construction, and other details are strong evidence for the conclusion claimed in the paper;

[2] Board coverage of different tasks in specific domains on assessing the ability of GPT-4 could encourage adopting LLMs as evaluation tools (e.g., reference-free evaluation), as more and more evidence has demonstrated a sufficient level of domain knowledge in LLMs;

[3] Strong experimental results of GPT-4 could motivate further research on developing more medical applications based on LLMs, which could be beneficial for the NLP community and society;


**Reasons To Reject:**

[1] Some of the findings in the paper could be found in the GPT-4 technical report (OpenAI, 2023);

[2] The accessibility of GPT-4 is weak and the findings in this paper may not be applicable to other LLMs (e.g., LLaMA), as there is still a performance gap between closed-source LLMs and open-source LLMs.

[3] As the overall experiments are based on remote LLMs (e.g., APIs) and the data (especially the licensed and medical ones) could be transited and stored in other servers, I personally think that this paper should include an ethics statement section to clarify any ethical concerns that might be raised by other readers.

References

OpenAI. 2023. GPT-4 Technical Report. arXiv.


**Reproducibility:**

3: Could reproduce the results with some difficulty. The settings of parameters are underspecified or subjectively determined; the training/evaluation data are not widely available.

**Reviewer Confidence:**

4: Quite sure. I tried to check the important points carefully. It's unlikely, though conceivable, that I missed something that should affect my ratings.

---

> ### Author Rebuttal · Authors · 2023-08-28
>
> Thank you for the positive review. We provide below rebuttal for the listed concerns:
>
> [1] Some of the findings in the paper could be found in the GPT-4 technical report (OpenAI, 2023);
>
> This paper focuses specifically on the radiology domain and answers the research question of how or whether general-domain GPT models (with no specialised training on radiology) can replace task/domain-specific models in the highly specialised radiology domain. Therefore, our paper complements the findings from the GPT-4 technical paper which focuses more on the general domain results.
>
> [2] The accessibility of GPT-4 is weak and the findings in this paper may not be applicable to other LLMs (e.g., LLaMA), as there is still a performance gap between closed-source LLMs and open-source LLMs.
>
> Thank you for the suggestion. In this paper, we focused on GPT-4 as it is the most capable and the best-performing LLM at the moment across many domains and we would like to establish what best we can do with a large language model in radiology. We acknowledge that the paper has not tested other LLMs  which are likely to yield inferior results compared with GPT-4, and we will explicitly address this in our limitation section for the next version.
>
>
> [3] As the overall experiments are based on remote LLMs (e.g., APIs) and the data (especially the licensed and medical ones) could be transited and stored in other servers, I personally think that this paper should include an ethics statement section to clarify any ethical concerns that might be raised by other readers.
>
> Thank you for the suggestion. We will add an ethical statement in the next version. In particular, we would like to assure the readers that we are using Azure Open AI services which have all the compliance requirements as any other Azure Services. Azure Open AI is HIPPA compliant and preserves data privacy and compliance of the medical data (eg. The data are not available to OpenAI). More details can be found in
> https://azure.microsoft.com/en-gb/resources/microsoft-azure-compliance-offerings/
> https://learn.microsoft.com/en-us/legal/cognitive-services/openai/data-privacy
> https://learn.microsoft.com/en-us/answers/questions/1245418/hipaa-compliance

---

### Official Review · Reviewer_Qzoi · 2023-08-04

**Soundness:** 4

**Excitement:**

4: Strong: This paper deepens the understanding of some phenomenon or lowers the barriers to an existing research direction.

**Paper Topic And Main Contributions:**

This paper conducted a close examination testing GPT-4's capability on understanding Radiology reports. The authors evaluated GPT-4 on a diverse range of common radiology tasks. For each task, they tried and evaluated different popular prompting strategies and compared the performances against SOTA domain specific systems. They also searched for and categorized errors made by GPT-4 using human judgement and proposed an error analysis framework. The paper presents good practices on prompting strategies across a variety of different tasks in radiology which can be a baseline for future research. The results of GPT-4 vs. SOTA models show that GPT-4 either outperforms or is on par with current SOTA radiology models. The qualitative analysis shows GPT-4 has a sufficient level of radiology knowledge and also indicates the limitations of GPT-4, directing future evaluation to move beyond mere metric scores.

**Reasons To Accept:**

- The paper is well written. The language is clear and easy to follow.
- The authors did exhaustive experiments on prompting GPT-4 across different radiology tasks with concrete examples and the results are well-structured.
- The qualitative

**Reasons To Reject:**

- The promptings look pretty baseline to me. It would be interesting to see if prompt tuning can improve the performance. Since this is a paper on testing the "boundaries" of GPT-4, the authors may include prompt tuning results for each prompt strategy to push for the best performance.

**Reproducibility:**

4: Could mostly reproduce the results, but there may be some variation because of sample variance or minor variations in their interpretation of the protocol or method.

**Reviewer Confidence:**

4: Quite sure. I tried to check the important points carefully. It's unlikely, though conceivable, that I missed something that should affect my ratings.

---

> ### Author Rebuttal · Authors · 2023-08-28
>
> Thank you for the positive review. We provide below rebuttal for the concern:
>
> "The promptings look pretty baseline to me. It would be interesting to see if prompt tuning can improve the performance. Since this is a paper on testing the "boundaries" of GPT-4, the authors may include prompt tuning results for each prompt strategy to push for the best performance. "
>
> Below is our rebuttal:
>
> Thanks for the suggestion. We did try a suite of prompting strategies including chain of thought, self-consistency, example selection/retrieval and iterative refinement. While we acknowledged in the limitation section that we did not exhaustively experimented with all recently proposed and more advanced prompting strategies (eg. tree-of-thought),  we would like to point out that GPT-4 with the current prompting strategies in the study already achieves near-ceiling performance in many tasks, and the findings from the paper should already represent what an average user can get out of using GPT models on these tasks.

---

### Official Review · Reviewer_rA88 · 2023-08-05

**Soundness:** 4

**Excitement:**

3: Ambivalent: It has merits (e.g., it reports state-of-the-art results, the idea is nice), but there are key weaknesses (e.g., it describes incremental work), and it can significantly benefit from another round of revision. However, I won't object to accepting it if my co-reviewers champion it.

**Paper Topic And Main Contributions:**

This paper conducted an evaluation of the recently released large language model GPT-4 by OpenAI in the context of radiology reports. It compares the model's performance with state-of-the-art radiology-specific models and investigates different prompting strategies. The results indicate that GPT-4 either performs as well as or surpasses specialized models. The study suggests that GPT-4 could serve as a unified foundation model for diverse domains, including radiology. Additionally, the paper proposed an evaluation framework and emphasized the importance of advanced prompting for certain tasks.

**Questions For The Authors:**

A: What is the cost of the test?
B: Is it possible that GPT-4 or ChatGPT have already seen that testing data during pre-training?

**Reasons To Accept:**

The paper conducted a comprehensive evaluation of GPT-4 on a wide range of radiology tasks, demonstrating its performance in language understanding and generation in the medical domain. The evaluation framework will also be publicly available to the community.

The authors compared GPT-4 with a list of state-of-the-art radiology-specific models in different tasks. The promising results show the potential of GPT-4 as a versatile foundation for various specialized tasks.

**Reasons To Reject:**

Overall, this work conducted a comprehensive evaluation of the existing GPT-4 on different tasks in the radiology domain, but it lacks novelty.

The GPT-4 model is quite expensive, especially when compared to open-source free models. This high cost may limit the usage and impact of the work.

Evaluation should include some other open-source LLMs like Vicuna and LLaMA for comparison, as they can also serve as foundation models across domains, and they are all available for free.

Qualitative evaluation by human experts may need for some open-ended tasks like summarization and generation.

**Reproducibility:**

5: Could easily reproduce the results.

**Reviewer Confidence:**

5: Positive that my evaluation is correct. I read the paper very carefully and I am very familiar with related work.

---

> ### Author Rebuttal · Authors · 2023-08-28
>
> Thank you for your review and questions.
>
> Regarding the concerns:
>
> 1. Evaluation should include some other open-source LLMs like Vicuna and LLaMA for comparison, as they can also serve as foundation models across domains, and they are all available for free.
>
> Thank you for the suggestion. In this paper, we focused on GPT-4 as it is the most capable and the best-performing LLM now across many domains and we would like to establish what best we can do with a large language model in radiology. It is currently out of the scope of the paper to test other LLMs and we will explicitly address this in our limitation section for the next version.
>
> 2. Qualitative evaluation by human experts may need for some open-ended tasks like summarization and generation.
>
> We agree that qualitative analysis is important for generation tasks. Indeed, we worked with a radiologist to provide qualitative analysis for a sample of the summarisation results in comparison with the human-written summaries. Appendix B.6 provides examples where the GPT-4 output is favoured by the radiologist with explanation. In addition, we are planning to conduct more nuanced qualitative evaluation addressing different aspects of the summary in the future, and we will add this discussion in the future direction section.
>
> Regarding the questions:
>
> A: What is the cost of the test?
>
> We used the Cognitive Service API for the study. The pricing is 0.06 USD per 1,000 tokens for prompt and 0.12 USD per 1,000 tokens. The actual cost depends on the task. For RadNLI with 361 samples, the cost is around 5 USD. The total cost of the test is around 5000 USD.  We acknowledge that the cost of GPT-4 is high at the moment. As such, the findings from our paper can save the costs from future researchers who want to investigate similar research questions.
>
> B: Is it possible that GPT-4 or ChatGPT have already seen that testing data during pre-training?
>
> Thanks for the question. We would like to point out that the majority of our testset comes from data with restricted access such as MIMIC. We will clarify this in the paper.

---

### Meta-Review · Area_Chair_SqbK · 2023-09-19

**Recommendation:** 4

**Metareview:**

This paper review evaluates the performance of the GPT-4 model in the context of radiology reports, comparing its performance against state-of-the-art radiology-specific models. The paper also investigates different prompting strategies and provides a detailed evaluation framework. The results demonstrate that GPT-4 either matches or surpasses the performance of specialized models, suggesting its potential as a versatile foundation for various specialized tasks. The paper also highlights the importance of advanced prompting for certain tasks.

All three reviewers acknowledge the value of the paper's extensive evaluations and analyses on GPT-4's performance in the radiology domain. They also appreciate the clarity and thoroughness of the paper. The reviewers raise concerns regarding the novelty of the work, the cost and accessibility of the GPT-4 model, and the need for comparison with other open-source large language models. They also suggest the inclusion of more advanced prompting strategies, qualitative evaluation by human experts, and an ethical statement section.

Overall, the paper presents valuable insights into the application of GPT-4 in the radiology domain, presenting findings that could inspire further research in the area, and the evaluation framework proposed could be beneficial to the community. It is recommended that the authors take into consideration the reviewers' suggestions for improvement in future work.

---

### Decision · Program_Chairs · 2023-10-07

**Decision:**

Accept-Main

**Comment:**

This paper review evaluates the performance of the GPT-4 model in the context of radiology reports, comparing its performance against state-of-the-art radiology-specific models. The paper also investigates different prompting strategies and provides a detailed evaluation framework. The results demonstrate that GPT-4 either matches or surpasses the performance of specialized models, suggesting its potential as a versatile foundation for various specialized tasks. The paper also highlights the importance of advanced prompting for certain tasks.

All three reviewers acknowledge the value of the paper's extensive evaluations and analyses on GPT-4's performance in the radiology domain. They also appreciate the clarity and thoroughness of the paper. The reviewers raise concerns regarding the novelty of the work, the cost and accessibility of the GPT-4 model, and the need for comparison with other open-source large language models. They also suggest the inclusion of more advanced prompting strategies, qualitative evaluation by human experts, and an ethical statement section.

Overall, the paper presents valuable insights into the application of GPT-4 in the radiology domain, presenting findings that could inspire further research in the area, and the evaluation framework proposed could be beneficial to the community. It is recommended that the authors take into consideration the reviewers' suggestions for improvement in future work.